# Multitask Classification and Segmentation for Cancer Diagnosis in Mammography

**Thi-Lam-Thuy Le**[1,2]                                         THILAMTHUY.LE@GE.COM

**Nicolas Thome**[1]                                           NICOLAS.THOME@CNAM.FR

**Sylvain Bernard**[2]                                     SYLVAIN.BERNARD@MED.GE.COM

**Vincent Bismuth**[2]                                        VINCENT.BISMUTH@GE.COM

**Fanny Patoureaux**[2]                                    FANNY.PATOUREAUX@GE.COM

[1] *MSDMA Team, CEDRIC Lab, Conservatoire National des Arts et Metiers, France*

[2] *Mammography Image Quality Team, GE Healthcare, France*

## 1. Introduction

Deep learning and Convolutional Neural Networks (ConvNets) (Krizhevsky et al., 2012; Durand et al., 2015) recently show outstanding performances for visual recognition. The representation learning capacity of ConvNets has also been successfully applied to medical image analysis and recognition in mammography (Huynh et al., 2016; Lévy and Jain, 2016; Geras et al., 2017; Kooi et al., 2017; Nikulin; Lotter et al., 2017).

A major challenge in the medical domain relates to the collection of a large amount of data with clean annotations, which is especially difficult due to the strong level of expertise required to perform data labelling. Regarding mammography image analysis for Computer-Aided Diagnosis (CAD), annotations can be very diverse, from global binary image labels (presence / absence of a cancer), *e.g.* DDSM (Heath et al., 2000), to finer-grained and pixel-level annotations (mass, calcification, both benign or malignant labels), to BI-RADS labeling, *e.g.* INbreast (Moreira et al., 2012).

In this work, we propose to leverage these heterogeneous but correlated forms of annotations to improve performances of deep ConvNets. To this end, we introduce a Multi-Task learning (MTL) scheme, which combines pixel-level segmentation and global image-level classification annotations. The proposed architecture is based on a Fully Convolutional Networks (FCN) Long et al. (2015), which enables efficient feature sharing between image regions and fast prediction. We evaluate our model on the DDSM database (Heath et al., 2000), with cancer classification and pixel segmentation with five classes. We show that the joint training is able to learn shared representations that are beneficial for both tasks. Our method can be seen as a generalization of approaches relying on detection annotations to pre-train deep model for a classification purpose, *e.g.* Nikulin; Lotter et al. (2017). We show that our joint training of classification and segmentation enables a better cooperation between tasks.

## 2. Proposed Multi-Task Classification and Segmentation Architecture

As shown in Figure 1, the proposed network is based on a FCN with a ResNet backbone (He et al., 2016) to extract local features. The FCN enables efficient feature computation in each region but also sharing computation from all regions in the whole image in a single forward pass. In addition, we can still process input images with high spatial resolution (e.g. $1152 \times 832$ in our experiments). We combine segmentation and classification, thus heterogeneous annotations can be leveraged to jointly optimize both tasks:

$$\mathcal{L}_{total} = \lambda \cdot \mathcal{L}_{cls} + (1 - \lambda) \cdot \mathcal{L}_{seg} \tag{1}$$

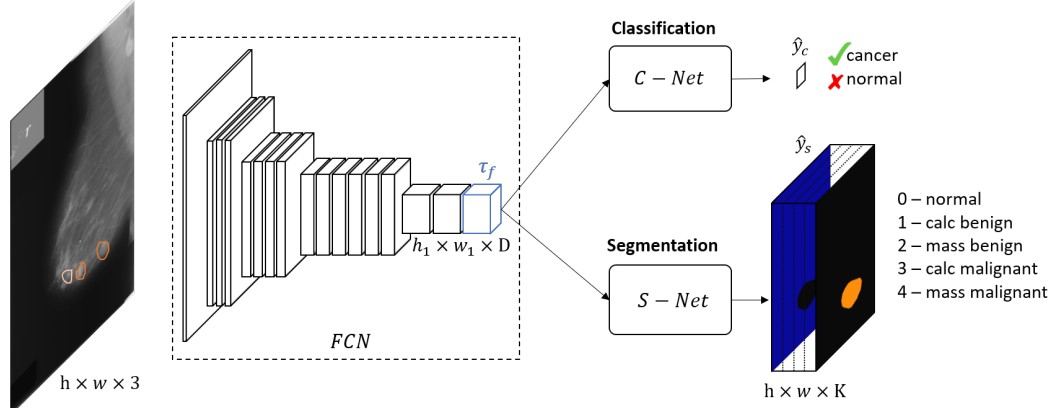

Figure 1: Overall architecture. It is based on a FCN to extract local features from the whole input images. Local features are then encoded by S-Net to yield lesion segmentation, and aggregated by C-Net to yield cancer classification.

**Segmentation Network (S-Net)** The segmentation network aims at classifying each pixel in the input image into a set of $K$ pre-defined classes. More precisely, from the output tensor $T_f$ of the backbone FCN, S-Net first consists in adding a transfer layer of $1 \times 1$ convolution to transform the output of the feature extraction network from large scale dataset to our K-class target dataset. Then, we perform an upsampling to create the semantic segmentation. On top of that, we define a weighted cross-entropy loss $\mathcal{L}_{seg}$ to address the class-imbalance issue ('healthy' regions are dominant compared to other lesion classes).

**Classification Network (C-Net)** The classification branch takes as input the shared tensor $T_f$ and outputs a single class label assessing the presence or absence of cancer in the input image. C-Net is composed of two steps. Firstly, local features are aggregated with a global average pooling (Lin et al., 2013) (GAP) to yield a single score per modality, over all regions in the input image. The second step consists in the last fully connected layer to get the final probability of cancer. We define the classification loss $\mathcal{L}_{cls}$ as a standard binary cross entropy as is common in classification tasks.

## 3. Results and Discussion

**Quantitative results** To highlight the relevance of our method, we perform classification and segmentation in three ways with the labeling scheme depicted in Figure 1: (1) train individually classification and segmentation models to get baseline performances on each task, (2) sequentially train the segmentation model and finetune local features for the classification task, and (3) jointly train both tasks. We evaluate segmentation and classification performances using respectively the mean Dice score over five classes and the Area Under Curve (AUC) of Receiver Operating Characteristic (ROC). As shown in Table 1, by pretraining the model on segmentation, our classification performance is slightly improved by $\sim 1$ pt to $AUC_{(2)} = 81.37\%$, compared to the pure classification with $AUC_{(1)} = 80.54\%$. As for our joint method, we achieved a significant gain both in segmentation and classification of $\sim 3.5$ pts ($meanDice_{(3)} = 38.28\%$) and $\sim 2.5$ pts ($AUC_{(3)} = 84.02\%$) respectively, compared to its sequential counterpart.

Table 1: Segmentation and classification performances on DDSM (Heath et al., 2000).

| Training strategy | Cls baseline$_{(1)}$ | Seg baseline$_{(1)}$ | Seg-cls sequential$_{(2)}$ | Seg-cls joint$_{(3)}$ |
|---|---|---|---|---|
| Seg perf (mean Dice) | - | 34.98 | 34.98 | **38.28** |
| Cls perf (AUC) | 80.54 | - | 81.37 | **84.02** |

**Qualitative results** Figure 2 illustrates classification and segmentation predictions for two images from DDSM, one normal and one cancer. In both cases, our joint method outperforms the sequential one for both classification and segmentation, and succeeds to capture lesions with highly precise localisation capability.

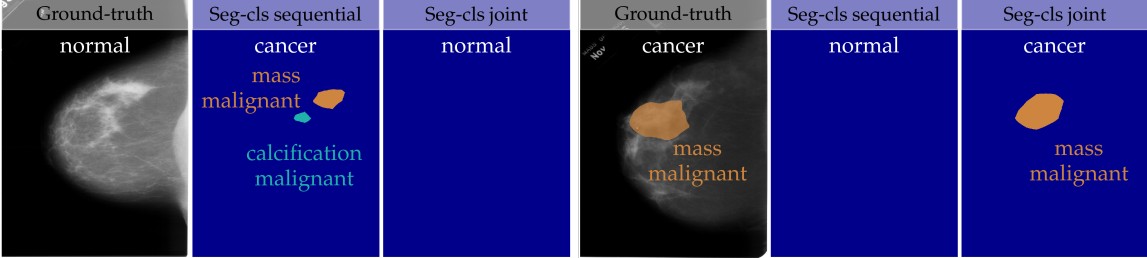

Figure 2: Segmentation and classification examples on DDSM (Heath et al., 2000). From left to right, the first three images show a normal case with ground-truth annotations and results from the segmentation-classification sequential and the segmentation-classification joint methods. The next three images show the same things for a cancer case.

## Conclusion

In this work, we proposed a multitask learning scheme which combines segmentation and classification for cancer diagnosis in mammography. The achieved performances have shown the effectiveness of our method on both recognition tasks, lesion segmentation and cancer classification. Future works include investigating more powerful models and leveraging heterogeneous datasets to improve predictive performances.

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

Yaroslav Nikulin. DREAM Challenge Yaroslav Nikulin (Therapixel) write up.

