# OpenReview forum: "Multitask Classification and Segmentation for Cancer Diagnosis in Mammography"
_MIDL.io/2019/Conference/Abstract — MIDL Abstract 2019_

### Official Review · AnonReviewer1 · 2019-04-29
**Joint segmentation and classification in mammography images**

**Rating:** 2
**Confidence:** 2

**Review:**

Summary:
A multi-task network has been proposed with shared initial layers, followed by separate sub-networks for pixel-wise segmentation (normal plus 4 other tissue types related to cancer) and image-level classification (cancer vs normal). Results demonstrate that joint learning of both tasks is better than learning them individually.

* No methodological novelty. It is well-known that joint learning helps when the tasks are related.
* Given the earlier point, I think that the experimental validation would have to be stronger to warrant acceptance. In this regard, I do not understand why sequential learning has been shown as one of the competing methods. If the data is available simultaneously, what would be the motivation for sequential learning?
* Further, it is unclear to me, exactly which weights are finetuned after the training of the segmentation task. The authors say that they 'finetune local features for the classification task', but which weight do they mean by this? I assume that the weights of the classification sub-network are finetuned, but the writing does not convey this clearly.
* Other instances of unclear writing:
        - what is meant by 'local features' extracted by the FCN?
        - what is meant by a 'transfer layer' of 1x1 convolution?

---

### Official Review · AnonReviewer2 · 2019-05-01
**Interesting method to combine various datasources**

**Rating:** 3
**Confidence:** 3

**Review:**

This paper is about both segmentation and classification of cancer in mammography. The goal is to leverage various datasources, with different labels available (tags or segmentation masks). The proposed solution is to modify a FCN (used for segmentation), and to add a classification branch in place of the deconvolution branch. Therefore, depending on the available label, the relevant branch will be selected. This will allow to train the same feature maps for both tasks.

Results shows an improvement for both task, when combining the sources (+3% dice, +3% AUC).

Suggestion:
Using tag information for segmentation is actually a Multiple Instance Learning setting: a positive tag means that at least one pixel should be positive, while a negative tag means that all pixels should be negative. There is previous literature in the field using that information, and could be compared to your method:
- Pathak, D., Krahenbuhl, P., & Darrell, T. (2015). Constrained convolutional neural networks for weakly supervised segmentation. In Proceedings of the IEEE international conference on computer vision (pp. 1796-1804).
- Kervadec, H., Dolz, J., Tang, M., Granger, E., Boykov, Y., & Ayed, I. B. (2019). Constrained-CNN losses for weakly supervised segmentation. Medical image analysis, 54, 88-99.

---

### Decision · Program_Chairs · 2019-05-06
**Acceptance Decision**

Accept